# Prototyping a Knowledge-Based System to Identify Botanical Extracts for Plant Health in Sub-Saharan Africa

**DOI:** 10.3390/plants10050896

**Published:** 2021-04-29

**Authors:** Pierre J. Silvie, Pierre Martin, Marianne Huchard, Priscilla Keip, Alain Gutierrez, Samira Sarter

**Affiliations:** 1PHIM Plant Health Institute, Montpellier University, IRD, CIRAD, INRAE, Institut Agro, F-34398 Montpellier, France; 2CIRAD, UPR AIDA, F-34398 Montpellier, France; pierre.martin@cirad.fr (P.M.); priscilla.keip@cirad.fr (P.K.); 3AIDA, Montpellier University, CIRAD, F-34398 Montpellier, France; 4LIRMM, Montpellier University, CNRS, F-34095 Montpellier, France; marianne.huchard@lirmm.fr (M.H.); alain.gutierrez@lirmm.fr (A.G.); 5CIRAD, UMR ISEM, F-34398 Montpellier, France; samira.sarter@cirad.fr; 6ISEM, Montpellier University, CNRS, EPHE, IRD, F-34398 Montpellier, France

**Keywords:** biopesticides, plant-based products, pesticidal plants, essential oils, crop protection, IPM, natural substances, knowledge management

## Abstract

Replacing synthetic pesticides and antimicrobials with plant-based extracts is a current alternative adopted by traditional and family farmers and many organic farming pioneers. A range of natural extracts are already being marketed for agricultural use, but many other plants are prepared and used empirically. A further range of plant species that could be effective in protecting different crops against pests and diseases in Africa could be culled from the large volume of knowledge available in the scientific literature. To meet this challenge, data on plant uses have been compiled in a knowledge base and a software prototype was developed to navigate this trove of information. The present paper introduces this so-called *Knomana* Knowledge-Based System, while providing outputs related to *Spodoptera frugiperda* and *Tuta absoluta,* two invasive insect species in Africa. In early October 2020, the knowledge base hosted data obtained from 342 documents. From these articles, 11,816 uses—experimental or applied by farmers—were identified in the plant health field. In total, 384 crop pest species are currently reported in the knowledge base, in addition to 1547 botanical species used for crop protection. Future prospects for applying this interdisciplinary output to applications under the One Health approach are presented.

## 1. Introduction

Crop production is hampered by the action of various organisms: competing plants, vertebrate pests (birds and mammals, including rodents), invertebrates (insects, mites, mollusks, nematodes) and diseases (fungi, viruses, bacteria, phytoplasma), some of which can be vectored by insects. These antagonistic organisms can develop on plants in the field, in greenhouses or postharvest, as well as on seeds or other stored food commodities.

Crop damage in the sub-Saharan region of Africa is caused by indigenous organisms or invasive exotic species [1]. *Helicoverpa armigera* (Hübner) (Lepidoptera: Noctuidae) is an indigenous species known to be a major cotton pest. Otherwise, the exotic species *Tuta absoluta* (Meyrick) (Lepidoptera: Gelechiidae) and *Spodoptera frugiperda* (J.E. Smith) (Lepidoptera: Noctuidae) are major pests of tomatoes and maize crops, respectively [2,3,4].

The use of synthetic pesticides (insecticides, acaricides, fungicides, nematicides, rodenticides, molluscicides) has also led to direct pest reductions, in turn reducing quantitative and qualitative crop yield losses. The compounds used in Africa are often formulated in commercial products manufactured abroad. This leads to high import and procurement costs for farmers with low purchasing power, particularly for farmers in landlocked countries without direct access to ports [5].

The impacts of major crop treatments on human health [6,7,8] and the environment (water pollution, reduction of pollinators, elimination of natural enemies of pests) are now increasingly well documented in sub-Saharan Africa [9], and in the tropics overall [10].

The loss of efficacy against target organisms through the inadvertent selection of pesticide-resistant individuals is another impact demonstrated in Africa [11,12] and globally, for example, neonicotinoid pesticides [13]. Resistance development renders these products ineffective, thereby forcing farmers to substitute them for other compounds with different modes of action.

The constraints and impacts associated with chemical pesticide usage have led to the development of strategies to reduce reliance on these substances. The integrated pest management (IPM) concept—now a paradigm—has been supported for over 60 years [14], yet with identified obstacles that hamper its adoption and widespread use [15,16]. More recently, the implementation of a variety of so-called agroecological practices has been recommended [17,18,19]. The biocontrol concept, for instance, promotes the use of four categories of biocontrol products, including natural substances. The adoption of other concepts or practices such as ecological engineering or agroecological engineering, habitat management, agroforestry and increased plant diversity, is highly recommended in suitable settings [20,21,22]. These practices are, however, complex to implement because they involve all actors in a given sector and area, whereas farmers are not always the landowners.

Use of plants or their extracts (aqueous, essential oils) is a conventional human health practice to ensure protection against harmful parasites or pathogens or alleviate the symptoms of various diseases [23], and in public health, to repel or eliminate vectors of organisms responsible for human diseases such as malaria [24,25,26]. There are also plant applications in the animal health field for terrestrial [27,28] or aquatic [29,30] farming, to directly control harmful organisms, enhance nutrition and thus the general health of animals and their disease resistance, while also curbing antibiotic usage.

This study focused specifically on the crop protection domain. Indeed, the use of plants to protect other plants is a practice regularly mentioned by farmers in surveys seeking to highlight their innovations [31,32,33,34] for either plants grown in the field or stored postharvest. Seeds are also protected in this way against soil pathogens. This crop protection strategy corresponds to the ‘substitution’ stage in the agroecological transition of food systems under the concept described by several authors [35,36,37].

Two types of substitution products can be used in this approach: substance formulations that are studied and certified like synthetic pesticides, or home-made preparations produced shortly before use.

### 1.1. Marketed Formulations

In industrialized countries, there are regulations and resources that in principle enable stakeholders to carry out all bioassays necessary for the certification of formulated products. Insufficient studies are however sometimes reported [38]. In the tests, impact assessments are carried out, including analyses of the effectiveness of the substance or formulation against target organisms, and measurement of unwanted effects against various categories of organisms (aquatic, terrestrial), including humans. In the public health field, essential oils are commonly used as repellents against adult mosquitoes [39]. There are also some examples, in the agriculture field, of products formulated and distributed on a commercial scale [40,41,42]. Natural pyrethrum extracts and neem-based formulations—respectively derived from cultivated *Tanacetum cinerariifolium* (Asteraceae) and *Azadirachta indica* (Meliaceae) plants—are widely used on different continents, despite certain identified shortcomings in terms of negative impacts or marketing [43,44,45,46].

### 1.2. Extemporaneous Preparations

The use of botanical extracts prepared immediately prior to their application to field crops is the second plant-based method often adopted by organic farmers [47,48]. Conventional plant-based protection of seeds and stored foods is also common practice [49,50].

Books or the proceedings of numerous meetings report various uses of plants for the protection of other plants [51,52,53,54]. Manuals are available for the preparation of these extracts in most countries worldwide [55,56,57]. The expertise outlined in these manuals may come from the collection of traditional knowledge [58,59]. The identification of plant species considered to be of interest for crop protection against pests and diseases is often based on properties observed in traditional medicine [60,61].

Historically, the advantages and limitations associated with the use of plants or their extracts have been widely described by authors from various continents in different media, ranging from journal articles to full-fledged books on the topic [62,63,64,65,66,67,68,69,70,71,72,73,74,75,76,77,78], particularly on essential oils [79,80,81,82,83,84,85,86,87]. Some problems are associated with non-commercial extracts, for example, the lack of regulations to provide for consistent preparation, quality control, efficacy testing, multiplication and conservation of plant species, unintended effects on non-target organisms.

Further knowledge has been acquired by the University of Greenwich and Kew Royal Botanic Gardens in two projects carried out in Africa: the Southern African Pesticidal Project (SAPP), as well as two EU-funded projects, i.e., African Dryland Alliance for Pesticidal Plant Technologies (ADAPPT) [88] and Optimising Pesticidal Plants: Technology Innovation, Outreach and Networks (OPTIONs) [89], carried out from 2009 to 2013. These projects paved the way for the different studies needed to enhance plant extract use while also stressing the need to preserve the diversity of traditionally collected plants [57,90].

Despite the plethora of studies published on the use of plants against all kinds of pests, it is still unclear why so few formulated products are available today, especially in Africa. The hypothesis we put forward in this article is that knowledge on these plant uses is relatively inaccessible and insufficiently disseminated to the various concerned actors: farmers, extension workers, NGO consultants, researchers, private entrepreneurs (processors), regulatory bodies and decision-makers. To fill this gap, a Knowledge-Based System (KBS) (e.g., [91]) for decision support, i.e., *Knomana* (short for knowledge management), has been developed to provide access to this knowledge for users.

*Knomana* aims to organize knowledge on the different uses of plants and their extracts so as to facilitate their implementation by the mentioned actors who have different research/application needs.

This paper presents the KBS structure, outlines the recorded knowledge on crop protection, and introduces a software prototype developed to mine this knowledge. Finally, it discusses the potential for its development to benefit other health domains, such as animal health, aquaculture, public health and environmental health, in line with the pluridisplinarity advocated under the One Health approach [92].

## 2. Materials and Methods

A project—funded by the Institut national de recherche pour l’agriculture, l’alimentation et l’environnement (INRAE) and the Centre de coopération internationale en recherche agronomique pour le développement (CIRAD) Transitions to Global Food Security (GloFoodS) Metaprogramme—was launched in 2017–2018 with the aim of developing KBS *Knomana*. It gave rise to the first stage of partnership building, notably in Cameroon and Burkina Faso, a literature search and the first data input [93]. It has also led to a collaboration with the Montpellier Laboratory of Computer Science, Robotics and Microelectronics (LIRMM) and the Institute of Evolutionary Science of Montpellier (ISEM) joint research unit.

The KBS is composed of three elements:The knowledge base (KB) presented in Section 2.1;The Conceptual Harvester (CH), which contains a set of methods to navigate and explore KB, as presented in Section 2.2;A reference library, which is implemented using Zotero software and contains all of the documents from which the data compiled in KB were extracted.

### 2.1. Description of the Knowledge Base (KB)

KB stores all KBS data. In a first step, for practical reasons, it has been implemented in Microsoft Excel^®^ as this software is readily available. KB currently comprises two types of dataset, i.e., ones that contain the collected data such as plant use descriptions or biopesticide formulations, and others that contain the dictionaries. The latter contain all information input in the observed data tabs, with each having an associated accepted value so as to standardize their value before data analysis. This standardization includes typing error correction (e.g., misspelled country names), as well as management of synonymous species names. In KB, the accepted name of a species corresponds to the accepted name provided by the Plants of the World Online website (www.plantsoftheworld.org, accessed on 1 March 2021) for plants and the Catalogue of Life website (www.catalogueoflife.org, accessed on 1 March 2021) for most pests. Additional information is associated with the standardized value of an entry, such as the categorization for each of the Linnaean taxonomy ranks for the accepted species name, or the continent for a country.

#### 2.1.1. Data Origin

The literature search began with a survey of published articles regarding research in French-speaking sub-Saharan African countries via the informal French-speaking *Plantes Pesticides d’Afrique* (PPAf) network that was set up in 2015 in 14 countries (Benin, Burkina Faso, Cameroon, Central African Republic, Côte d’Ivoire, Democratic Republic of the Congo, Equatorial Guinea, Madagascar, Mali, Mauritania, Niger, Nigeria, Senegal and Togo). Their feedback also led to the retrieval of other publications. This first literature survey helped spot references not reported in conventional reference databases. A complementary search in reference databases such as the Web of Science was conducted with the keywords ‘pesticidal plants’ and ‘Africa’. A focused approach was taken by targeting special topics, such as plants grown, particularly in organic farming, plants requiring protection (cabbage, cocoa, cotton) or domain (stored foods, e.g., [94,95]) or by targeting particular pests, such as the invasive alien insect species *Spodoptera frugiperda* and *Tuta absoluta*.

As the volume of references was substantial, not all articles or article reviews have been entered yet. A reference management software (Zotero) pools the analyzed articles in pdf format. An operating license granted to the contributors ensures the protection of these data. The literature search has also begun on a broader geographical scale, including other continents, mainly through the reading of reviews (e.g., [96,97]). All data included in the articles have been manually logged.

#### 2.1.2. Structure of the Microsoft^®^ Excel File

In the observed data category, data are organized as a double entry table, i.e., a data type per column and a dataset per row. Each datum is entered as it is presented in the literature to prevent any typing errors or misinterpretation. Any entry adjustment is solved later, using the associated dictionary before analysis.

Latin names of the plant species are placed in rows in the first column, as presented in the queried literature. The following columns contain the attributes associated with each row, information from bibliographical consultations and the definition in relation to the consultation chronology. Additional attributes/columns can thus be added on the right side of the matrix. When compounds of botanical origin are used without a record of the source host plant of the extraction, ‘not determined’ is indicated in the first column.

Columns related to plant species used for crop protection include the vernacular name in the language indicated in another column. The botanical family and different organs of the plant used are also listed, as well as the active ingredient extraction method (or the commercial name if it is a ready-made commercial formulation) and the usage form (e.g., powder, oil, essential oil). One column indicates whether a biochemical analysis was carried out and, if so, additional columns detail the chemical identification techniques used (e.g., GC/MS), along with the major chemical compounds reported in the article. One column is devoted to the active ingredients used alone. When using bioguidance, the biological tests are often carried out on a fraction of an extract or a single molecule.

The same applies to the geographical origins of the plants, which are detailed if this information is available, in order to take potential chemotypes into account. One column then refers to the country from which the plant originates, even if it has been tested in another country, another column concerns the region or a different geographical entity and a final column refers to the sampling site.

Organisms are specified by their Latin name or the common name indicated in the article. Dictionaries can be helpful for specifying these terms. One column specifies the field of application (field protection, seeds, stored foods). These two columns thus define the ‘protected system’. The next three columns specify some characteristics of the organisms targeted by the plants used, the type of organism (insect, mite, pathogen), its Latin name and the targeted development stage (larva, adult). When the tests involved natural enemies, especially beneficial insects such as predators or parasitoids, their names are also listed in these columns. Two groups of columns then describe the application procedures, including laboratory bioassays and field experiments. In the case of laboratory tests, it is specified whether the extract is used alone or combined with other components, the application method is indicated in another column (e.g., contact, ingestion, leaf application, inhalation, imbibition), while the applied dose is given in the unit reported in the article. For extracts applied in the field, the columns indicate whether the extract is used alone or in combination, the usage form (powder, liquid, mixture, emulsion, etc.), the type of preparation (extemporaneous, ready-made formulation), the applied mixture volume, the area treated, the dose applied in the unit reported in the article (L. ha^−1^, kg/L), as well as the applied per-ha dose when there is a single active ingredient.

Interactions between the plant used and the target organism—characterizing the desired effects—are noted in three columns. The first concerns the types of effect studied on the physiology (toxicity and biocidal effect) or behavior (repellent effect, possible attractiveness, anti-appetence, anti-feeding, anti-oviposition), but without reaching the degree of precision reported by some authors, in the case of repellent effects [98]. The second column specifies the activities or variables measured, which are closely linked to the type of organism targeted. The third column gives the overall effect or interaction observed. This last column may, for instance, indicate an average mortality rate mentioned in the article. Another column assigns an overall quality to the collected information. An article is considered to be of reliable quality if a biochemical analysis was carried out and was well described.

To filter documents according to various criteria, the bibliographical references were then detailed by separating the authors’ names, publication year, document or publication title, journal, volume and page number, while also specifying the nature of the document (article, book, grey literature). The name of the journal can be another criterion to select the information with reliable quality, when it is a referenced scientific journal.

When a review article leads to the logging of several other references, the author names and publication year are listed in a special column. A column indicates the DOI numbers. Traceability is achieved by storing all documents in pdf format in a Zotero library. The code assigned to each reference in the Zotero library is given in a column in the general matrix.

This general matrix has been supplemented over time, with columns added on the right side. These columns report the indicated effects in a more quantitative way than in the articles, with a broad range of measured variables related to the test methods, e.g., mortality at different observation dates, median lethal concentrations and doses (LC_50_, LD_50_), cycle time, weight reduction, fungal growth inhibition, etc.

One line in the matrix represents ‘one knowledge’: experimental or non-experimental (i.e., already adopted in the field) use of a given plant extract applied at a given dose (when specified) to protect a given plant from the harmful effects of an antagonistic target organism (pests, diseases of various origins).

Publications mentioning unintended effects are also considered. They describe the effects of the extracts on non-target organisms, e.g., soil organisms (earthworms), aquatic organisms (daphnia), beneficial arthropods (natural enemies, pollinators) or plants (extract phytotoxicity).

Review articles enable logging of a lot of information at once, without reaching the level of detail of the original cited articles. An analysis and redistribution of tables of the original publication (a review or not) is required to isolate each use per line.

For example, if an extract has been tested at several doses against several target species, each dose and species is represented in one line in the matrix.

A degree of reliability is attributed to the analyzed publication. Any publication that does not identify the chemical composition of the applied extracts is kept.

#### 2.1.3. Ontology and System Architecture

Relationships between the concepts (Microsoft^®^ Excel file attributed titles) can be represented as a formal structure or ontology [99]. Ontology can be referred to ‘as a specification of a representational vocabulary for a shared domain of discourse’ [100] or to ‘a formal and explicit specification of a shared conceptualization’ [101]. The adopted knowledge representation is based on a ternary relationship between the plant used for protection (biopesticide), the target organism (pest, disease or, in the current construction, beneficial organism) and the protected system (pre or postharvest crop) (Appendix A).

There are two ways of using knowledge in KB. The purpose of the *navigation* is to identify existing pieces of knowledge, while the *exploration* aims to create new pieces of knowledge or identify knowledge gaps. The computer methods to carry out these two tasks—via the construction of dedicated algorithms—are designed within the general framework of a system consisting of two interrelated components, the knowledge base (KB) and the Conceptual Harvester, itself made up of four elements (Appendix A), i.e., the query builder, the knowledge extractor, the reasoning tool that analyses the data and the knowledge visualization tool. The first and last of these elements facilitate user interaction.

### 2.2. Conceptual Harvester

The Conceptual Harvester (CH) is software that contains a set of methods to navigate and explore KB [102]. Exploratory conceptual navigation methods are used to enable users formulating general and potentially imprecise queries without prior knowledge of the large mass of data compiled in KB [103]. These methods are based on the building of conceptual classifications using formal concept analysis (FCA)—a symbolic artificial intelligence method based on the lattice structure [104,105]. FCA is already applied to various domains (e.g., life science, software engineering) [106,107,108] as it enables efficient data classification, association rule extraction and searches for frequent knowledge patterns. Conceptual classifications are similar to biological classifications, i.e., data are classified in groups, subgroups, sub-subgroups, etc., in a hierarchical manner such as the Linnaean taxonomy ranks (family, genus, species, etc.). By using the conceptual classification, a group includes a set of entities (e.g., *Spodoptera frugiperda*, *Tuta absoluta*) and a set of characteristics shared by these entities (e.g., invasive species, eat tomato). Contrary to a biological classification that has a tree structure, a conceptual classification is a more general partial order classification, i.e., one group can have two super-groups, while a species cannot belong to several genera. As a hypothetical example, one can have the group of toxic plants and the group of medicinal plants, and a sub-group of both would be toxic medicinal plants, such as *Digitalis purpurea* L. The group of toxic medicinal plants has two super-groups: the group of toxic plants and the group of medicinal plants. Lattice structures are particular partial orders (e.g., see Appendix A).

In the case of relational and multi-table data, such as in KB, use of the relational concept analysis (RCA) method, i.e., an extension of FCA, is preferred [107,109,110,111]. With RCA, entities are described by characteristics, as well as by relations to other entities from the same category (e.g., a pest *eats* another pest) or from different categories (a plant *repels/keeps away* a pest). In this analysis framework, different conceptual classifications (e.g., one for plants and another for pests) are also produced that rely on shared characteristics and shared relationships. The building of a classification for an observed dataset using RCA requires an ontology associated with a category (see Appendix A).

One dataset issue that arises is related to the presence of the ‘sp.’ or ‘spp.’ abbreviations, which correspond to the absence of the species name (only the genus is mentioned), which is considered as an indeterminate data value. A description model for taking this indeterminate information into account in FCA and RCA has been proposed [112].

## 3. Results

In early October 2020, the knowledge base (KB) hosted data extracted from 342 documents of various origins (publications in scientific journals, conference proceedings, academy bulletins, etc.), written in English or French, and dated from 1957 to 2020. Of these articles, 11,816 uses have been identified in the plant health field, whether experimental or applied by farmers.

### 3.1. Content of the Knowledge Base (Plant Health)

#### 3.1.1. Protected Systems

In the plant health field, crops may be protected in different periods—during field cultivation, possibly including seed treatment before sowing, and during postharvest storage, with leaf applications or with plants or extracts placed in storage structures. The association between the plant to be protected and its immediate environment constitutes the protected system. Different protection areas can be categorized as follows:*Crop protection in the field*: this applies to all field crops.*Seeds*: this concerns the preservation of plant organs to be sown the following crop season and pre-sowing treatments, if any.*Stored grain*: this concerns postharvest grain storage, often in granaries, which is very important in African dryland areas to ensure a food supply for the population during the ‘lean season’.*Foods*: this concerns the food crop postharvest period. Plant extracts are used to prevent spoilage of plant-derived food by contaminants such as toxin-producing microorganisms.

Seventy cultivated plant species are listed in the KB. The extracts of some of them are also indicated as being suitable for protecting other plants, e.g., *Carica papaya* (Caricaceae), *Anacardium occidentale* (Anacardiaceae) or Lamiaceae species.

A table of uses—experimental or applied by farmers—may be drawn up by navigating through KB by type of crop to be protected.

Tomato is a vegetable that is frequently mentioned as requiring protection. Depending on the location and preparation/application method, plants help protect crops against fungi (*Aspergillus niger*, *Athelia rolfsii*, *Fusarium graminearum*, *F. oxysporum*, *F. poae*, *Phytophthora infestans*), insects (*Helicoverpa armigera*, *Tuta absoluta)* and mites (*Tetranychus evansi*). Table 1 shows the botanical species used to protect tomato. The botanical families most represented in KB are Apocynaceae, Asteraceae, Fabaceae, Lamiaceae, Myrtaceae and Piperaceae.

Another example concerns bean (Table 2). Fewer families and species are listed in KB compared to tomato.

Cowpea is a crop that is often mentioned as requiring protection because its stored grains are attacked by many insects and diseases.

Regarding the protection of stored foods, Table 3 presents plant species listed in KB that have been tested or applied to control cowpea weevils (*Callosobruchus maculatus* (F.), Coleoptera: Chrysomelidae, Bruchinae) in Africa.

Asteraceae, Lamiaceae and Poaceae (genus *Cymbopogon*) are the botanical families most reported in KB.

#### 3.1.2. Targeted Organisms

In total, 384 crop pest species are currently listed in KB.

These organisms belong to a broad range of taxa:Arthropods, such as insects or mites, some insects that may be plant disease vectors, such as the aphid *Brevicoryne brassicae* (L.) or the whitefly *Bemisia tabaci* (Gennadius). Insect pests of grain stored in cowpea or maize granaries are also present, such as the weevils *Callosobruchus maculatus* [113] and *Caryedon serratus* (Olivier);Phytopathogenic nematodes such as *Meloidogyne incognita* or *M. javanica*;Pathogenic microorganisms causing fungal diseases (*Alternaria solani, Aspergillus flavus*, *Fusarium oxysporum*). Bacterial diseases are only represented by *Xanthomonas campestris* pv. *malvacearum* [96].

#### 3.1.3. Pesticidal Plant Species

In total, 1547 plant species are used in whole form or in different extract forms (aqueous, alcoholic, essential oils).

The botanical families with the largest number of species listed in KB are Lamiaceae (297), Asteraceae (274), Fabaceae (243), Apiaceae (79), Myrtaceae (79), Euphorbiaceae (76), Rutaceae (65), Annonaceae (60), Apocynaceae (58), Meliaceae (54), Rubiaceae (54), Solanaceae (52) and Malvaceae (43).

Table 4 shows the ranking of plant species currently in KB with more than 50 records (=lines) (see Appendix A for the full data). All species present with the same number of lines are ranked similarly. The maximum rank is 79. Neem (*A. indica*) is the most recorded species (710 lines), followed by *Ocimum gratissimum* (Lamiaceae) (306 lines). Concerning the case of 237 lines, the botanical species was not noted in the publication, so only one (or more) compound was tested in the search reported in KB.

Searches in the available literature revealed non-crop species in addition to those already reported in the ADAPPT and OPTIONs project factsheets. Plants such as *Lantana camara* (Verbenaceae), *Hyptis suaveolens* (Lamiaceae), *Calotropis procera* (Apocynaceae) and *Tephrosia vogelii* (Fabaceae) could thus potentially be very interesting [114,115,116]. They represent a readily available reservoir of plants considered to be weeds, as also reported for *Tithonia diversifolia* (Asteraceae) [117].

### 3.2. Results for Two Invasive Species (T. absoluta and S. frugiperda)

Regarding invasive alien insects on the African continent, navigation has the advantage of facilitating identification of the plant species that have been used in the original ranges of these insects. The case of *Tuta absoluta* (Lepidoptera: Gelechiidae) and *Spodoptera frugiperda* (Lepidoptera: Noctuidae), which were quite recently identified in Africa but whose global dissemination is ongoing, is outlined below.

For *T. absoluta*, KB lists species belonging mainly to the families Asteraceae (eight species), Lamiaceae (seven species), Piperaceae (six species), Rutaceae and Salicacae (four species each) (Table 5).

Regarding *S. frugiperda*, Table 6 presents the search results obtained on all Lepidoptera species of the *Spodoptera* genus, associating—according to a ternary relationship—the different species of this genus with the plant species used and the protected crops. A greater number of species have been tested with regard to maize protection, especially in South America, the original range of *S. frugiperda*. The plants to be protected against *Spodoptera littoralis* (Boisduval) are cotton, okra (*Abelmoschus esculentus*), cabbage and tomato, while castor beans were protected against *Spodoptera litura* (F.) attacks.

In the case of the reported binary relationships (*Spodoptera* species, plant used, without the protected plant mentioned), a much higher number of plant species were tested (Appendix A): 279 species belonging to 62 botanical families, for *S. littoralis*, 205 species belonging to 48 families for *S. frugiperda*, 80 species (28 families) for *S. litura*, 6 species (4 families) for *Spodoptera exigua* (Hübner), 2 species (one family) for *S. eridania* (Stoll) and only one for *S. ornithogalli* (Guenée) and *S. exempta* (F.). In the case of *S. frugiperda*, some of the plant species mentioned are cultivated. The total number of plant species recorded in the knowledge base (279) is higher than the 70 species reported in a recent review [117]. Amongst those reported, *L. camara* and *T. vogelii* appear to have little impact on *S. frugiperda* [118].

Regarding *S. frugiperda*, families with the most species tested include Asteraceae and Meliaceae (29 species each), Annonaceae (20 species), Fabaceae (13 species), Lamiaceae (11 species), Euphorbiaceae (9 species) and Myrtacaeae (8 species) (Appendix A).

In the case of *S. littoralis*, the most represented families are Lamiaceae (70 species), Asteraceae (44 species), Apiaceae (24 species), Fabaceae (21 species) and Apocynaceae (9 species). For *S. litura*, 14 Lamiaceae species are listed in KB, while 11 Asteraceae, 8 Rutaceae, 7 Solanaceae and 5 Fabaceae are other families with fewer species represented.

The range of listed families generally corresponds to the original geographical distribution of the three *Spodoptera* species, i.e., the Americas, Europe, the Mediterranean Basin and Asia.

## 4. Discussion

The first two points addressed concern KB building process and development of the associated computer analysis methods that facilitated exploitation of the huge volumes of data available from ethnobotanical surveys and published academic studies. The potential of developing KBS for other health fields is discussed.

### 4.1. Knowledge Base (KB)

Navigating in KB is currently performed using queries according to different entries that correspond to the key descriptive ontology concepts (Appendix A): (i) cultivated plant species to be protected; (ii) harmful organisms that attack them (pests, diseases) at sowing, on field plants, or postharvest; (iii) plants used and their geographical origins. However, other filters may also be applied to collect information on the doses used, their impacts, to only select review articles, etc.

Baseline publications that have been logged may also be queried via the DOI numbers and the coding of bibliographical references from a Zotero library. Species cropped for human consumption, such as papaya and cashew, have also been focused on. Their use for other purposes would warrant broader dissemination.

With the development of computer analysis methods, KB also enables the production of novel knowledge, corresponding to KB exploration, that may be proposed for experiments, e.g., concerning the use of repellent plant species as part of an insect pest management (IPM) push-pull strategy [119,120]. It enables identification of plant genera or species to be tested against exotic pests for which knowledge is available on the use of extracts in their area of origin, as in the case for *T. absoluta.*

However, KB still has some limitations at this building stage. First, only information that has been logged is available, so not all crops grown in the sub-Saharan region are present among the protected systems reported in KB. Some terrestrial target organisms are poorly represented (Gasteropods) or not at all (Myriapods), while some species can cause significant damage to leaves or seedlings. Extensions regarding the use of algae and soil biofumigation could be made for plant extracts. Other substances could be added, such as biostimulants, along with definitions of new features specific to these uses.

The collected information quality could be questioned because, as some authors have pointed out, the research carried out is often insufficiently described in many publications [121]. In particular, there is a lack of chemical characterization frequently reported in the case of plant species proven effective against *S. frugiperda* [117].

For example, the Latin name, chemotype or precise geographical area are not always provided in publications, although this information is important for explaining any potential variability in the results obtained in a given region, or according to the seasons [122,123,124,125]. Voucher samples are seldom deposited in collections.

Chemical analyses of essential oil extracts are often carried out, but this is seldom true for aqueous extracts that are readily used by some communities. This raises the issue of the certification by researchers of the effectiveness of extract preparations in NGO guides and manuals, which are actually often the result of the observation of traditional practices. Regarding ready-made formulations, the official certification process automatically calls for in-depth studies on extract compositions.

KB could be enhanced by adding links with other knowledge bases focused on the chemical composition of plants. One criterion for selecting a candidate species in a given country could be not taxonomic (family, genus or species) but instead chemical, i.e., a species whose major compounds are close to those identified in known extracts. Some compounds may also be production residues (byproducts), e.g., seeds of species of the Annonaceae family that contain acetogenins [126,127]. Wash water from *Boscia senegalensis* (Capparaceae) seeds could also be interesting to study with regard to allelopathic effects, linked to the glucosinate compound content [128], in addition to the insecticidal effects noted with different organs of this plant [129]. The mode of action of the extracts could also be a criterion added to KB [40].

In terms of formulations, a follow-up of the literature and additions could be included under the relevant KB category so as to take advances achieved with nanoencapsulation methods into account [130,131,132,133].

In the publications queried to build KB, the efficacy parameters measured were highly variable, i.e., heterogeneous in the units presented (LD_50_, LD_90_, etc.) in relation to the various methods used, thereby complicating any efforts to conduct a meta-analysis.

In the current KB, environmental health is approached through the unintentional effects on non-target organisms, such as soil organisms (earthworms), aquatic organisms (daphnia), beneficial arthropods (pollinators, predators (insects, mites)) and parasitoid insects of crop pests. A comparison of selected candidate species with botanical species that can affect fish [134] would be a way to warn users of the usage risks. In this health field, the volume of recorded publications on unintentional effects is tiny (1% of current knowledge in the entire KB) and this could be improved by new inputs from available publications. For example, there are studies that could be added that deal with the effects of extracts on bees [135] or natural enemies such as parasitoid [136,137] or predatory insects [138,139,140]. Other effects such as extract phytotoxicity could also be recorded, not only with regard to the crops to be protected but also to other plants [141]. These elements will be essential, particularly for the development of formulations which must comply with homologation standards. Allelopathic effects during cultivation of plants with a pesticidal effect could also be of interest, particularly with regard to weeds, with the aim of reducing glyphosate usage.

### 4.2. Knowledge Engineering

*Knomana* data and expert needs have been a source of knowledge for engineering research questions.to enable navigation and exploration. Methods and algorithms have been developed and tested based on KB. Although developed for KB, they are general and apply to similar datasets and research questions. Solutions for applying FCA/RCA in data models including ternary relationships have been proposed [142]. They consider several encodings of ternary relationships, including solutions using binary relationships. These different solutions enable the extraction of various forms of relevant knowledge patterns. An on-demand algorithm for building conceptual structures for RCA has been defined [143]. This algorithm gradually builds the RCA concept lattices: starting from one concept of one entity category (e.g., plants), on-demand, it is able to build either subconcepts (subgroups) or superconcepts (supergroups) in the same lattice, or neighbor concepts in another entity category (e.g., pests) on the basis of the relationships and relational attributes. Solutions for dealing with indeterminate data introduced with sp. and spp. have been introduced in [112]. These solutions apply to RCA as well as to other FCA extensions. Combining these extensions enhances the knowledge conveyed to plant experts. FCA and RCA have been applied to KB excerpts at different stages in its building process. In [142], the uses of six plants selected by experts for their recognized qualities against *Aspergillus* were analyzed to answer the question: ‘Knowing the recognized benefits of *Hyptis suaveolens* in the protection of *Arachis hypogaea* against *Aspergillus parasiticus*, which other plants could alternatively be used?’. The RCA-based classification and plant grouping readily revealed that *Ocimum gratissimum* could replace *Hyptis suaveolens* to protect *Arachis hypogaea* against *Aspergillus ochraceus* and *Aspergillus flavus*. In [112], a subset limited to three insect species of the *Spodoptera* genus and indeterminate species information on 30 pesticidal plants and six protected crops was investigated. This enabled qualitative comparison of the different methods by establishing the shape of the extracted knowledge patterns, while highlighting potential scalability problems. Moreover, in the case of a fungal disease (*Fusarium oxysporum*), an algorithm has enabled identification of *Lantana camara*, a plant species of interest for controlling *F. oxysporum* in Burkina Faso. Indeed, this species is used in Benin to control the same fungal pathogen and is found in Burkina Faso where it is used against another *Fusarium* species, i.e., *F. solani* [102]. This newly generated knowledge could be the focus of future experiments in Burkina Faso.

### 4.3. Identifying Multipurpose Species: A Promising Approach

Prior to the implementation of certain complex solutions involving agroecological practices, many different applications of plants and their extracts may already be identified to promote their use. Ready-made formulations are preferable to reduce preparation times that often discourage farmers, while ensuring extract quality and facilitating development of plant production chains. A hypothesis could be put forward that more economic outlets would be available for multipurpose species. Examples of such species with multiple uses exist. *Tephrosia vogelii* [123] and *Mucuna pruriens* (Fabaceae) [144] can be used to improve soil fertility and produce extracts. A recent review highlights the benefits of certain cover crops used in conservation agriculture for soil pest biocontrol [145]. Plant species identified as companion plants could also be selected [146]. Species of the Apiaceae, Asteracae and Lamiaceae families have been found to serve as a floral food source for natural enemies in a crop association (habitat manipulation), while also serving to produce extracts [147]. This multi-use approach seems very promising in the plant health field.

Furthermore, as pointed out in the introduction, uses in areas other than plant health could be worthwhile to explore. The repellent effects of essential oils are known in the public health field [148,149,150]. In animal health, particularly to enhance aquaculture sustainability, there is growing interest in herbal therapy for reducing antibiotic usage, enhancing fish resistance to diseases and improving growth and feed efficiency [151]. The effects of plant-derived compounds on fish parasites have been reviewed [152].

Positive effects of herbal supplementation on fish growth performance have often been reported. For instance, *Cinnamosma zeylanicum* (Lauraceae) was found to have a positive effect on Nile tilapia growth and feed utilization [153]. Medicinal herbs have been described to act as immunostimulants enhancing the non-specific defense mechanisms of fish and shellfish and decreasing their mortality after experimental infection with pathogens [154]. It has been shown that furanones from the alga *Delisea pulchra* are able to protect brine shrimp (*Artemia franciscana*) against *Vibrio harveyi*, *V. campbellii* and *V. parahaemolyticus* infection [155,156]. Moreover, it has been reported that these compounds are able to protect rainbow trout from vibriosis [157], but they have also presented some toxicity to these fish. This clearly indicates that the catalogue of quorum sensing inhibitor compounds has to be extended and, interestingly, it has been revealed that plant extracts are promising potential sources of such original compounds [158].

In addition to the cultivation of plants for human and animal consumption (cereals, vegetables, legumes, tubers, etc.) and clothing (cotton), there are many other uses: dye plants, plants that provide plant cover for direct sowing, nectariferous plants that favor the maintenance of parasitoid or pollinating insects, plants that are environmental bioindicators, etc. [147,159,160].

To group and record the various uses in a pragmatic way, several matrices could be developed according to each use. A KB extension is therefore being planned to identify the diverse range of services offered by plant species.

Moreover, with a view to cropping (without synthetic pesticide treatments) plants of interest identified according to the multipurpose criterion, trophic chains that concern them will have to be assessed, which means identifying their pests and diseases. Indeed, plants with pesticidal or antimicrobial effects can also be attacked by pests or diseases, e.g., *Moringa oleifera* (Moringaceae) in Niger [161]. The search for phytophagous insects attacking *Lantana camara* has even been voluntarily undertaken with a view to biological control of this invasive plant [162]. Another ontology linking all species in these trophic chains will be needed, similar to that defined for cereal stem boring insects [163].

Finally, the implementation of KB using Microsoft Excel^®^ was a first step in the development of the Knowledge-Based System. The motivation to use the latter raised the availability of this software on the computers of the partners who entered the data and the ease to aggregate additional types of information to describe plant use. Effectively, the enlargement to other protected systems (animal, human, etc.) was conducive to the adding of specific columns. Now, the structure of KB is stable enough to consider the development of the end-user application (EEA), in which the final knowledge container (database, triplestore, or other) will be identified in relation to the information technology solution that will be adopted to develop EEA. The development of CH is currently under progress. The first version of the Knowledge-Based System will be developed for smartphone and designed for farmers.

## 5. Conclusions

The development of a software program to provide any user with easy access to knowledge is a keystone of the research program carried out by the group of researchers presented here. Building a Knowledge-Based System for spotting plants or their extracts for plant health is one step of this program. It will be further enhanced through the development of a KB extension to include the broad and diverse range of services offered by plant species. Additional thematic categories should also be added to, for instance, describe trophic chains between organisms and other plant uses in particular fields (animal health, dye plants, service plants including cover crops for direct seed or ‘push-pull’ plants, biofuel plants, plants for paralyzing fish, food plants, plants useful for soap making, nectar plants). Knowledge management reaching beyond the disciplinary boundaries of plant health will facilitate development of the transdisciplinarity advocated by the One Health approach.

## 6. Patents

The KB software structure and a first version of the knowledge base (Usage des plantes à effet pesticide, antimicrobien, antiparasitaire, antibiotique/Use of plants with pesticidal, antimicrobial, antiparasitic and antibiotic effects) were registered in 2019 with the European Agency for the Protection of Programs under numbers 122264 and 122779. An operational license is currently granted only to contributors of the first version.

## Figures and Tables

**Table 1 plants-10-00896-t001:** Examples in KB of plant genera and species used for the protection of tomato (*Solanum lycopersicon*).

Botanical Family	Species	Botanical Family	Species
Amaranthaceae	*Dysphania ambrosioides*	Lythraceae	*Lawsonia inermis*
Amaryllidaceae	*Allium cepa*	Meliaceae	*Azadirachta indica*
	*Allium sativum*		*Entandrophragma angolense*
Apiaceae	*Coriandrum sativum*		*Melia azedarach*
	*Eryngium foetidum*		*Trichilia pallida*
Apocynaceae	*Allamanda cathartica*	Moraceae	*Ficus elastica*
	*Calotropis gigantea*	Musaceae	*Musa* sp.
	*Calotropis procera*	Myrtaceae	*Callistemon citrinus*
	*Vincetoxicum canescens*		*Eucalyptus camaldulensis*
	*Vincetoxicum fuscatum*		*Eucalyptus saligna*
	*Vincetoxicum parviflorum*		*Eucalyptus tereticornis*
Asteraceae	*Acanthostyles buniifolius*		*Eugenia egensis*
	*Acmella oleracea*		*Syzygium aromaticum*
	*Ageratum conyzoides*	Nitrariaceae	*Peganum harmala*
	*Ageratum houstonianum*	Nyctaginaceae	*Bougainvillea glabra*
	*Artemisia absinthium*	Oxalidaceae	*Oxalis barrelieri*
	*Artemisia annua*	Papaveraceae	*Argemone mexicana*
	*Artemisia cina*	Piperaceae	*Piper aduncum*
	*Artemisia vulgaris*		*Piper amalago*
	*Bidens pilosa*		*Piper augustum*
	*Calendula officinalis*		*Piper glabratum*
	*Eclipta prostrata*		*Piper mikanianum*
	*Emilia coccinea*		*Piper mollicomum*
	*Erigeron floribundus*	Poaceae	*Cymbopogon citratus*
	*Guizotia abyssinica*		*Digitaria eriantha*
	*Tagetes erecta*	Podocarpaceae	*Podocarpus milanjianus*
Bignoniaceae	*Spathodea campanulata*	Primulaceae	*Clavija weberbaueri*
Brassicaceae	*Brassica rapa*	Rosaceae	*Rosa damascena*
Capparaceae	*Crateva religiosa*	Rutaceae	*Citrus × aurantium*
Clusiaceae	*Garcinia smeathmanii*		*Citrus limon*
Commelinaceae	*Commelina benghalensis*		*Citrus reticulata*
Cupressaceae	*Tetraclinis articulata*		*Citrus sinensis*
Dilleniaceae	*Curatella americana*		*Clausena anisata*
Dipterocarpaceae	*Shorea robusta*	Salicaceae	*Banara guianensis*
Euphorbiaceae	*Euphorbia hirta*	Salicaceae	*Banara nitida*
	*Jatropha curcas*		*Mayna parvifolia*
	*Ricinus communis*		*Ryania speciosa*
Fabaceae	*Bauhinia variegata*	Sapindaceae	*Deinbollia saligna*
	*Copaifera duckei*	Sapotaceae	*Argania spinosa*
	*Ononis natrix*		*Madhuca longifolia*
	*Pongamia pinnata*	Simmondsiaceae	*Simmondsia chinensis*
	*Sesbania bispinosa*	Siparunaceae	*Siparuna poeppigii*
	*Tephrosia vogelii*	Solanaceae	*Nicotiana* sp.
Geraniaceae	*Pelargonium zonale*		*Solanum delagoense*
Lamiaceae	*Ajuga chamaepitys*	Tropaeolaceae	*Tropaeolum majus*
	*Mentha spicata*	Urticaceae	*Urtica dioica*
	*Ocimum basilicum*	Verbenaceae	*Lippia javanica*
	*Ocimum gratissimum*		*Lippia multiflora*
	*Thymbra capitata*	Zingiberaceae	*Elettaria cardamomum*
	*Thymus vulgaris*		
	*Zataria multiflora*		

**Table 2 plants-10-00896-t002:** Examples in KB of plant genera and species and parts of the plant used for the protection of bean (*Phaseolus vulgaris*).

Botanical Family	Species	Parts of the Plant Used
Amaranthaceae	*Chenopodium opulifolium*	All, dried leaves
	*Dysphania ambrosioides*	Leaves
Amaryllidaceae	*Allium sativum*	Cloves
Asphodelaceae	*Aloe* spp.	All
Asteraceae	*Tithonia diversifolia*	Dried leaves
	*Vernonia amygdalina*	Dried leaves
Dryopteridaceae	*Dryopteris filix-mas*	Leaves
Fabaceae	*Gliricidia sepium*	Fresh or dried leaves, seeds
	*Senna siamea*	All
	*Tephrosia vogelii*	All
Lamiaceae	*Ocimum gratissimum*	All, leaves
Meliaceae	*Azadirachta indica*	All
	*Melia azedarach*	Dried leaves
Myrtaceae	*Callistemon viminalis*	Leaves
	*Eucalyptus* spp.	All
Solanaceae	*Capsicum annuum*	Not indicated
	*Capsicum* spp.	All, fresh fruits
	*Nicotiana tabacum*	All
Urticaceae	*Urtica dioica*	Leaves

**Table 3 plants-10-00896-t003:** Examples of plant species tested or applied to control *Callosobruchus maculatus* infestation of cowpea (*Vigna unguiculata*) stocks in Africa.

Botanical Family	Species	Botanical Family	Species
Amaranthaceae	*Dysphania ambrosioides*	Meliaceae	*Azadirachta indica*
Annonaceae	*Annona muricata*		*Khaya senegalensis*
	*Annona senegalensis*		*Melia azedarach*
	*Monodora myristica*	Moraceae	*Ficus exasperata*
	*Xylopia aethiopica*	Moringaceae	*Moringa oleifera*
Apiaceae	*Foeniculum vulgare*	Myrtaceae	*Callistemon rigidus*
Apocynaceae	*Pergularia daemia*		*Corymbia citriodora*
Asparagaceae	*Dracaena arborea*		*Eucalyptus saligna*
Asteraceae	*Blumea oloptera*		*Eucalyptus staigeriana*
	*Blumea viscosa*	Opiliaceae	*Opilia amentacea*
	*Helianthus annuus*	Poaceae	*Cymbopogon citratus*
	*Tagetes minuta*		*Cymbopogon flexuosus*
	*Tithonia diversifolia*		*Cymbopogon giganteus*
	*Vernonia amygdalina*		*Cymbopogon nardus*
Boraginaceae	*Heliotropium indicum*		*Cymbopogon schoenanthus*
Capparaceae	*Boscia senegalensis*		*Cymbopogon winterianus*
	*Crateva religiosa*	Polygalaceae	*Securidaca longipedunculata*
Caricaceae	*Carica papaya*	Rutaceae	*Clausena anisata*
Combretaceae	*Combretum imberbe*	Solanaceae	*Capsicum annuum*
	*Combretum micranthum*		*Capsicum* spp.
Cucurbitaceae	*Momordica charantia*		*Nicotiana tabacum*
Euphorbiaceae	*Euphorbia lateriflora*	Verbenaceae	*Lantana camara*
	*Spirostachys africana*		*Lippia javanica*
Fabaceae	*Chamaecrista nigricans*		*Lippia multiflora*
	*Gliricidia sepium*		*Lippia rugosa*
	*Tephrosia densiflora*	Zingiberaceae	*Alpinia calcarata*
	*Tephrosia vogelii*		
Lamiaceae	*Hyptis spicigera*		
	*Hyptis suaveolens*		
	*Ocimum americanum*		
	*Ocimum basilicum*		
	*Ocimum gratissimum*		
	*Plectranthus glandulosus*		
	*Tetradenia multiflora*		

**Table 4 plants-10-00896-t004:** Ranking of plant species used in plant health (extract from Appendix A) according to their occurrence (=number of lines = knowledge) in KB.

Scheme 710.	Botanical Family	Number of Lines	Rank
*Azadirachta indica*	Meliaceae	710	79
*Ocimum gratissimum*	Lamiaceae	306	78
Not indicated	Not indicated	237	77
*Dysphania ambrosioides*	Amaranthaceae	211	76
*Allium sativum*	Amaryllidaceae	194	75
*Lantana camara*	Verbenaceae	178	74
*Cymbopogon citratus*	Poaceae	164	73
*Tephrosia vogelii*	Fabaceae	156	72
*Ocimum basilicum*	Lamiaceae	142	71
*Carica papaya*	Caricaceae	130	70
*Callistemon citrinus*	Myrtaceae	120	69
*Melia azedarach*	Meliaceae	120	69
*Citrus sinensis*	Rutaceae	114	68
*Ageratum conyzoides*	Asteraceae	91	67
*Melia volkensii*	Meliaceae	89	66
*Rosmarinus officinalis*	Lamiaceae	86	65
*Senna crotalarioides*	Fabaceae	85	64
*Citrus limon*	Rutaceae	79	63
*Nicotiana tabacum*	Solanaceae	73	62
*Thymus vulgaris*	Lamiaceae	71	61
*Syzygium aromaticum*	Myrtaceae	69	60
*Tithonia diversifolia*	Asteraceae	69	60
*Moringa oleifera*	Moringaceae	67	59
*Euphorbia hirta*	Euphorbiaceae	66	58
*Xylopia aethiopica*	Annonaceae	66	58
*Citrus reticulata*	Rutaceae	64	57
*Capsicum annuum*	Solanaceae	63	56
*Ricinus communis*	Euphorbiaceae	63	56
*Monodora myristica*	Annonaceae	62	55
*Erigeron floribundus*	Asteraceae	61	54
*Euphorbia lateriflora*	Euphorbiaceae	61	54
*Pimpinella anisum*	Apiaceae	60	53
*Eucalyptus camaldulensis*	Myrtaceae	59	52
*Foeniculum vulgare*	Apiaceae	57	51
*Bidens pilosa*	Asteraceae	55	50
*Vernonia amygdalina*	Asteraceae	54	49
*Hyptis spicigera*	Lamiaceae	53	48
*Boscia senegalensis*	Capparaceae	52	47
*Mentha × piperita*	Lamiaceae	52	47
*Cuminum cyminum*	Apiaceae	51	46
*Eclipta prostrata*	Asteraceae	51	46
*Eucalyptus tereticornis*	Myrtaceae	51	46
*Commelina benghalensis*	Commelinaceae	50	45
*Emilia coccinea*	Asteraceae	50	45
*Oxalis barrelieri*	Oxalidaceae	50	45
*Podocarpus milanjianus*	Podocarpaceae	50	45

**Table 5 plants-10-00896-t005:** Plant species reported in KB as used against the tomato pest *Tuta absoluta*.

Botanical Family	Genus or Species	Botanical Family	Genus or Species
Amaranthaceae	*Dysphania ambrosioides*	Meliaceae	*Azadirachta indica*
Amaryllidaceae	*Allium cepa*		*Melia azedarach*
	*Allium sativum*		*Trichilia pallida*
Apiaceae	*Coriandrum sativum*	Myrtaceae	*Eucalyptus camaldulensis*
Apocynaceae	*Allamanda cathartica*		*Eugenia egensis*
Asteraceae	*Acanthostyles buniifolius*		*Syzygium aromaticum*
	*Acmella oleracea*	Nitrariaceae	*Peganum harmala*
	*Ageratum conyzoides*	Nyctaginaceae	*Bougainvillea glabra*
	*Artemisia absinthium*	Papaveraceae	*Argemone mexicana*
	*Artemisia annua*	Piperaceae	*Piper aduncum*
	*Artemisia cina*		*Piper amalago*
	*Artemisia vulgaris*		*Piper augustum*
	*Calendula officinalis*		*Piper glabratum*
Bignoniaceae	*Spathodea campanulata*		*Piper mikanianum*
Capparaceae	*Crateva religiosa*		*Piper mollicomum*
Cupressaceae	*Tetraclinis articulata*	Poaceae	*Cymbopogon citratus*
Dilleniaceae	*Curatella americana*	Primulaceae	*Clavija weberbaueri*
Euphorbiaceae	*Jatropha curcas*	Rosaceae	*Rosa damascena*
	*Ricinus communis*	Rutaceae	*Citrus × aurantium*
Fabaceae	*Bauhinia variegata*		*Citrus limon*
	*Copaifera duckei*		*Citrus reticulata*
	*Ononis natrix*		*Citrus sinensis*
Geraniaceae	*Pelargonium zonale*	Salicaceae	*Banara guianensis*
Lamiaceae	*Ajuga chamaepitys*		*Banara nitida*
	*Mentha spicata*		*Mayna parvifolia*
	*Ocimum basilicum*		*Ryania speciosa*
	*Ocimum gratissimum*	Sapotaceae	*Argania spinosa*
	*Thymbra capitata*	Simmondsiaceae	*Simmondsia chinensis*
	*Thymus vulgaris*	Siparunaceae	*Siparuna poeppigii*
	*Zataria multiflora*	Solanaceae	*Nicotiana* sp.
Lythraceae	*Lawsonia inermis*	Tropaeolaceae	*Tropaeolum majus*
		Urticaceae	*Urtica dioica*
		Zingiberaceae	*Elettaria cardamomum*

**Table 6 plants-10-00896-t006:** Plant species reported in KB in relation to the *Spodoptera* genus and host plants to be protected.

Pesticidal Plant	Targeted *Spodoptera* Species	Plant to Be Protected
*Azadirachta indica*	*Spodoptera frugiperda*	*Zea mays*
	*Spodoptera littoralis*	*Abelmoschus esculentus*
		*Brassica oleracea*
*Croton macrostachyus*	*Spodoptera frugiperda*	*Zea mays*
*Curcuma longa*		
*Cymbopogon martini*		
*Dysphania ambrosioides*		
*Eucalyptus globulus*		
*Jatropha curcas*		
*Juniperus communis*		
*Lantana camara*		
*Limnanthes alba*		
*Melaleuca alternifolia*		
*Millettia ferruginea*		
*Nicotiana tabacum*		
*Phytolacca dodecandra*		
*Schinus molle*		
*Syzygium aromaticum*		
*Trichilia casaretti*		
*Trichilia catigua*		
*Trichilia claussenii*		
*Trichilia elegans*		
*Trichilia pallens*		
*Trichilia pallida*		
*Carica papaya*	*Spodoptera littoralis*	*Abelmoschus esculentus*
		*Brassica oleracea*
*Dioscorea dumetorum*	*Spodoptera littoralis*	*Gossypium hirsutum*
*Vincetoxicum canescens*		*Solanum lycopersicum*
*Vincetoxicum fuscatum*		
*Vincetoxicum parviflorum*		
*Wollastonia dentata*	*Spodoptera litura*	*Ricinus communis*
*Capsicum* spp.	*Spodoptera* spp.	*Zea mays*

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
