# Peer review of "Prototyping a Knowledge-Based System to Identify Botanical Extracts for Plant Health in Sub-Saharan Africa"

_plants, 2021, doi:10.3390/plants10050896_

Round 1

Reviewer 1 Report

The article “Towards a knowledge-based system to spot botanical extracts for plant health in sub-Saharan Africa '' is devoted to the so-called Knomana knowledge-based system, that contains information on plant extracts, which can be used for protection of agricultural plants. Of course, searching for non-toxic nature-born compounds for plant protection is of great interest, and the compilation and analysis of data available is a very important task. But I am perplexed concerning the information that was reported in the article presented. In the first instance, when I read a manuscript, I try to find the essential scientific idea of the article. What is the essential idea of this article? I read about partnership with Cameroon, Burkina Faso and scientific institutes in France, the history of the development of Knomana. This is Microsoft Excel file (?) accompanied with Conceptual Harvester (?) software, that, in my opinion, most likely is an algorithm composed using tools Excel Macro. Is it a fully-featured database? WS Excel file is MS Excel file, it isn't a database as NCBI, it cannot be used by other programs, etc.
Thus, in this article authors wrote how they collect and organize information for their own private use. In this MS Excel file users can find the list of species, which were reported as “protective” for the target species, or defense from the target pathogen.
Trustability of data collected is questioning. Thus, Table 4 contains information on 237 cases, when botanical species was not noted in the publication.
Using this Knomana authors found that, for example, Neem (A. indica) is the most recorded in literature species (710 lines), followed by 394 Ocimum gratissimum (Lamiaceae) (306 lines). Families with the most species tested on activity against Spodoptera frugiperda include Asteraceae and Meliaceae (29 species each), Annonaceae (20 species), Fabaceae (13 species), Lamiaceae (11 species), Euphorbiaceae 434 (9 species) and Myrtacaeae (8 species).  This data can be a part of literature review, but this article is not literature review, isn’t it?
In my opinion, the article should be rejected or transferred to another reviewer for new analysis

Reviewer 2 Report

The spirit of the paper is interesting and the information gathered are very useful to have a general vision of the problem, even if it is a particular geographical area of Africa. What does not convince me is the title of the work that does not completely represent the contents of the manuscript, in my opinion it would be useful to insert the word “review” and specify the main cultures taken into consideration.

Reviewer 3 Report

Dear authors

it is a very nice work. Please pay attention to some syntax errors. Also the text is not justified in some parts.

Reviewer 4 Report

The manuscript entitled “Towards a knowledge-based system to spot botanical extracts for plant health in sub-Saharan Africa” by Silvie et al. developed a knowledge-based system for plants which is a prototype.

Minor comments

  • The abstract conveys the message the article intends to convey. However, authors need to rewrite the abstract where they could not cover the all-present findings of this study.
  • The introduction part is lengthy with so many references, please reduce the references where possible.
  • Line 54-56 is looking out of context.
  • Line 62 seems to be repeating the above para.
  • As mentioned in the result section (line 363), rice, maize, and cowpea crop species requiring protection but later in the manuscript, nothing is discussed about rice and maize except cowpea.
  • In table 1, KB of plant genera and species that used for protection of tomato are mentioned but missed the data like protection against which microorganisms/mode of action/which part of the plant used/which plant extract used, if data available. Same for the remaining tables.
  • The software developed in the present study is looking like preliminary and needs to add more data because it is restricted to a smaller number of plant species.
  • Cross-check the reference section, they are not in format.
  • Overall, the language is satisfactory. But I would suggest thorough proofreading for rectifying grammatical and usage errors if any.

The content in the current version has included most of the relevant literature available but there a few critical suggestions to be made. I would recommend the publication of this manuscript after addressing minor changes.
